## Research Article

suicide; suicidal ideation; suicide attempt; eastern Ethiopia

**Corresponding author:**
Kemal Aliye;
Email: kemalaliye69@gmail.com

# Suicidal ideation and attempts among adults in eastern Ethiopia: A community-based cross-sectional study

Kemal Aliye[1] , Kabtamu Nigussie[1], Mekdes Demissie[1], Tadesse Misgana[1], Tilahun Abdeta[1], Daniel Alemu[1], Gari Hunduma[1], Dejene Tesfaye[1], Tilahun Betemariam[1], Dawud Wedaje[1], Abdulsalam Assefa[2], Mandaras Tariku[1], Henock Asfaw[1], Abdulkarim Amano[1] and Fethia Mohammed[1]

[1]Psychiatry, Haramaya University College of Health and Medical Sciences, Ethiopia and [2]Medicine, Haramaya University College of Health Sciences, Ethiopia

## Abstract

Suicide is a significant global mental health issue and a leading cause of death, with over half a million lives lost annually. The majority of suicide deaths occur in low- and middle-income countries (LMICs), yet there are limited data on suicidal behavior in these regions, particularly in Ethiopia. Given the rising trends of mental health issues worldwide and the alarming rate of suicide in LMICs, this research addresses a critical gap in knowledge regarding suicidal behavior in Eastern Ethiopia, which is essential to inform local public health strategies. Therefore, the study aimed to assess the prevalence and associated factors of suicidal ideation and attempts among adults in the Kersa, Haramaya and Harar Health and Demographic Surveillance System in Eastern Ethiopia in 2022. A community-based cross-sectional study was conducted among 1,411 participants selected using a multistage sampling technique. Binary logistic regression was employed to identify factors associated with suicidal ideation and attempts.

The findings revealed that 9.8% and 6.2% of participants reported suicidal ideation and attempts, respectively. A history of mental illness [adjusted odds ratio (AOR) = 6.82, 95% confidence interval (CI): 4.63–10.05] and khat use (AOR = 2.34, 95% CI: 1.48–3.69) were factors significantly associated with suicidal ideation. Similarly, rural residence (AOR = 4.32, 95% CI: 2.17–7.58), a history of mental illness (AOR = 6.02, 95% CI: 3.78–9.60) and khat use (AOR = 2.23, 95% CI: 1.29–3.85) were strongly associated with suicide attempts (p < 0.05). The study highlights that nearly one in 10 individuals in the community experienced suicidal ideation or attempts, underscoring the urgent need for attention to these mental health concerns. In conclusion, suicidal ideation and attempts are prevalent in Eastern Ethiopia and are significantly associated with mental illness, khat use and rural residence. Early screening, detection and management of suicidal behavior at the community level are recommended to address this pressing public health issue.

## Impact statement

Suicide is the major overlooked public health problem in many low- and middle-income countries, including Ethiopia. Although most suicide death occurs in this region, data to guide prevention efforts are limited. This study provides important evidence about the extent of suicidal ideations and attempts among adults in Eastern Ethiopia and identifies specific groups who may be at higher risk. By offering population-based data from three major surveillance sites, the finding contributes important information that can help local health systems, policymakers and community-based programs.

The findings indicating that nearly one in 10 adults experienced suicidal ideation demonstrate that suicide is not a rare or hidden issue in these communities. Identifying history of mental illness, khat use and rural residence as key associated factors highlight opportunities for targeted intervention. These results suggest that strengthening mental health services, reducing stigma and integrating mental health screening into existing community health programs could significantly improve early detection and prevention.

This study impacts extends beyond local context. It adds to global mental health knowledge by providing evidence from a region where suicide research is scarce. The findings underscore the need for culturally sensitive, community-based mental health strategies in similar low-resource settings. By bringing attention to the magnitude of suicidal ideations and attempts in Eastern Ethiopia, this finding supports advocacy for investment in mental health infrastructure and policy development.

Overall, the study emphasizes that suicide prevention should be prioritized as part of broader public health efforts.



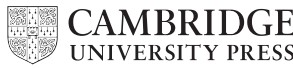

## Introduction

Suicidal ideation and attempts are significant public health concerns worldwide, contributing to the global burden of morbidity and mortality. According to the World Health Organization (WHO), over 700,000 people die by suicide annually, making it one of the leading causes of death globally, particularly among individuals aged 15–29 years (WHO, 2021). Suicidal behavior, which encompasses suicidal ideation, planning and attempts, is often associated with a complex interplay of psychological, social and biological factors (Franklin et al., 2017). Despite its global prevalence, low- and middle-income countries (LMICs), including Ethiopia, bear a disproportionate burden of suicidal behavior due to limited mental health resources, stigma and inadequate awareness (Mars et al., 2014).

In Ethiopia, mental health issues, including suicidal behavior, are underreported and poorly understood, particularly in rural and underserved regions. Studies conducted in different parts of the country have reported varying prevalence rates of suicidal ideation and attempts, with higher rates observed among individuals with mental health disorders, substance use and those experiencing socioeconomic hardships (Gelaye et al., 2013). However, data on suicidal behavior among adults in Eastern Ethiopia remain scarce, despite the region's unique cultural, economic and social dynamics that may influence mental health outcomes.

Eastern Ethiopia is characterized by high levels of poverty, unemployment and limited access to healthcare services, which may exacerbate the risk of suicidal behavior (CSA and ICF, 2016). Additionally, cultural stigma surrounding mental health issues often discourages individuals from seeking help, further complicating the prevention and management of suicidal behavior (Hailemariam et al., 2015). Understanding the prevalence and associated factors of suicidal ideation and attempts in this region is critical for developing targeted interventions and improving mental health services.

This study aimed to assess the prevalence of suicidal ideation and attempts among adults in Eastern Ethiopia and identify associated risk factors. By providing community-based data, this research seeks to inform public health strategies and policies aimed at reducing the burden of suicidal behavior in the region. The findings will contribute to the growing body of literature on mental health in LMICs and highlight the need for culturally sensitive interventions to address this pressing issue.

## Methods

### Study design and setting

This community-based cross-sectional study was conducted from October 1 to October 30, 2022, at the Kersa, Haramaya and Harar Health and Demographic Surveillance Sites (HDSS) in Eastern Ethiopia. The Harar HDSS, located in the Harari Regional State, is an urban site situated 510 km from Addis Ababa. The region is divided into six urban and three rural districts, further subdivided into 19 urban and 17 rural kebeles (the smallest administrative unit in Ethiopia, with an average population of 5,000). According to the 2007 national census, the region had a population of 183,344, with 54.2% residing in urban areas and 45.8% in rural areas. The population growth rates from 2007 to 2010 were 2.0% in urban areas and 3.3% in rural areas (Assefa et al., 2016). Haramaya HDSS was established in the Haramaya district, located in the East Hararghe zone of the Oromia region, 500 km from Addis Ababa and

18 km west of Harar. It is bordered on the south by Kurfa Chele district, on the west by Kersa, on the north by Dire Dawa, on the east by Kombolcha and on the southeast by the Harari Region. Its administrative center, Haramaya town, is located at 42° 3′ E, 9° 26′ N, at an altitude of 1,980 m above sea level. East Hararghe, which is one of 20 zones in the Oromia region, has 20 districts, including the Haramaya district. Haramaya has 34 rural kebeles (subdistricts) and three urban kebeles in Haramaya town. The data from the district office show that the total number of residents in the district is 310,310. The catchment area of Haramaya HDSS includes 12 of the 34 rural kebeles in the Haramaya district.

The Kersa HDSS is located in the Kersa sistrict of the Oromia regional state, bordered by Bedeno, Meta, Dire Dawa, Haramaya and Kurfa Chele districts. The district's elevation ranges from 1,400 to 3,200 m above sea level. There are 35 rural subdistricts (kebeles) and three small-town kebeles. According to the 2007 national census, the district has a total population of 172,626 of whom 6.9% are urban residents. The Kersa HDSS covers 24 of the 38 kebeles, with four health centers and 10 health posts. There are 18 elementary, two secondary, one preparatory and two religious' schools in the HDSS area, as well as 134 mosques, eight churches and six farmers' training stations. Most inhabitants are farmers, with a minority working in small trades, government posts or casual laborers. These sites were selected to represent diverse urban and rural populations, providing a comprehensive understanding of suicidal behavior in the region (Assefa et al., 2016).

### Source population, study population and inclusion and exclusion criteria

All residents living in Kersa, Haramaya and Harar HDSS were the source population. All residents living in Kersa, Haramaya and Harar HDSS of selected kebeles, adults aged 18 years and above, who had been residing in the district for at least six months were included in this study. Participants who were unable to communicate or who had serious medical illnesses were excluded from the study.

### Sample size determination and sampling procedures

The sample size was estimated using a single-population proportion formula [$p = (Z_{\alpha/2})^2 p (1 - p)/d^2$]. Therefore, the sample size was based on the assumption that the prevalence of mental distress would be about 27.9% (Jordans et al., 2018), with a design effect of 1.5 (due to the multistage sampling of study participants) and 10% of nonresponse rate, the total sample size of 1,416 participants were considered for this study because the suicidal behaviors was a part of this mega project we used the same sample size. This sample size was also used for assessing suicidal behaviors, as it was part of a larger project.

A multistage probability sampling technique was employed. There were 24 kebeles in Kersa, 12 in Haramaya and 12 in Harar DHSS. From the total number of kebeles at each site, six, three and three sample kebeles were selected using the lottery method. The total sample size was allocated proportionally to each kebele. Each sampling unit (household) was selected by systematic random sampling. The first study unit was randomly selected, and if there was more than one study participant (individuals aged 18 or above) in a household, a simple random sampling method was used. If an eligible respondent was not found in the selected household, the next household was selected for study (Figure 1).

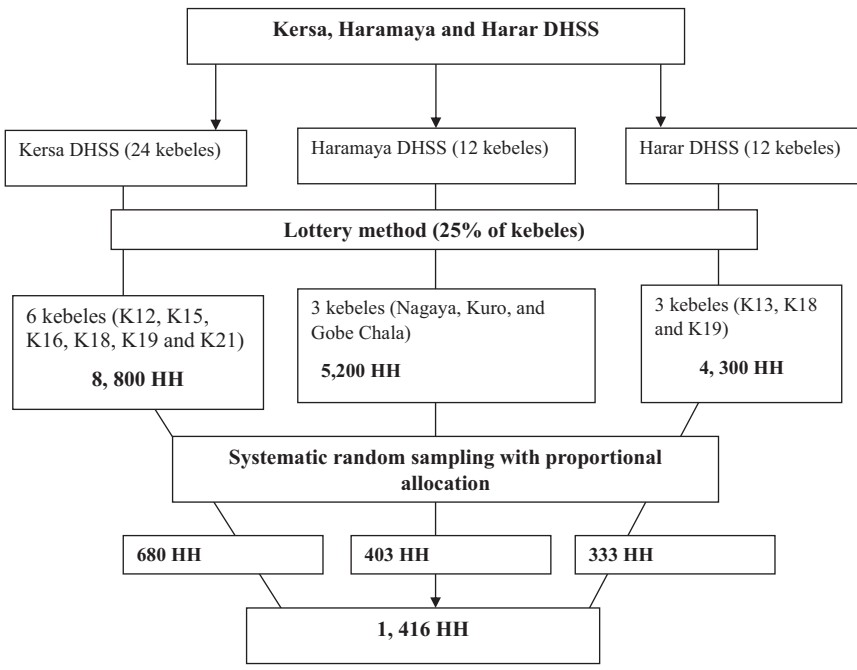

**Figure 1.** Schematic presentation of sampling procedure for assessing suicidal ideation and attempt in Kersa, Haramaya and Harar Health and Demographic Surveillance Sites, Eastern Ethiopia.

## Data collection instruments and measurements

Data were collected using face-to-face interviews, and structured questionnaires were used to collect all required data on the socio-demographic, economic and clinical characteristics of the participants such as religion, age, sex, marital status, level of education, occupation, residence, living arrangement, previous history of mental illness, history chronic medical illness and family history of mental illness.

Suicidal ideation and attempts were assessed using items adapted from the World Mental Health (WMH) survey initiative version of the WHO Composite International Diagnostic Interview, in which suicide was studied and validated in Ethiopia, both in clinical and community settings, with internal consistency Cronbach's alpha = 0.97. If the participant's answered "yes" for the question: have you ever seriously thought about committing suicide? He/she had suicidal ideation and also if the participants response "yes" for the item question: have you ever attempt suicide? He/she had a suicide attempt (Miller et al., 2000). An affirmative response to this item was coded as presence of lifetime suicidal ideation and attempt.

Smoking and substance involvement screening test [Alcohol, Smoking and Substance Involvement Screening Test (ASSIST)]: Psychoactive substance use was assessed using the ASSIST. The tool was developed by the international group of substance use researchers of the WHO, with a sensitivity and specificity of 97and 90, respectively. It is a brief screening questionnaire used to determine participants' use of psychoactive substances. It is mainly used to assess the psychoactive substance use of lifetime and current users (Heslop et al., 2013).

Social support was assessed by Oslo Social Support (Oslo-3): The Oslo-3, referred to as 14-point Social Support Scale, is a brief and economic instrument to assess the level of social support. It can be integrated into larger research projects, such as population-based studies, without significantly enhancing the efforts of both the researchers and participants. Oslo-3 consists of only three items asking for the number of close confidants, the sense of concern from other people and the relationship with neighbors, with a focus on the accessibility of practical help. Individuals who scored 12–14, 9–11 and 3–8 were had strong, moderate and poor social support, respectively, using 3-items Oslo Social Support Scale (Kocalevent et al., 2018).

Stressful life events screening questionnaires (SLESQ) were used to measure stressful life events,13 item self-report measures for nontreatment-seeking samples that assess lifetime exposure to traumatic events (Goodman et al., 1998).

Data were collected by nine trained bachelor's degrees in psychiatry nursing and supervised by two trained supervisors with MSc in public health. All interviews for data collection were completed by adhering to the COVID-19 prevention protocol, and data were collected using the Amharic or Afan Oromo versions.

## Data quality assurance and management

The structured questionnaires were first prepared in English, translated into Amharic and Afan Oromo, and then re-translated into English by a language expert to check its consistency. To ensure data quality, we adapted standardized tools for the research objectives. To evaluate the acceptability and applicability of procedure and tools, a pretest was performed on 5% of sample size before one-week actual data collection took place on the kebeles that were not included in the samples from each site. To maintain the completeness and consistency of the questionnaires, supervisors and investigators closely supervised the data collectors during the data-collection process.

## Statistical analysis

Data were cleaned and checked for completeness, coded, entered into the computer using EPI info version 3.1 and imported into STATA version 14.0. Categorical variables were expressed as

frequencies and percentages. Bivariable logistic regression analyses were used to assess the association between dependent variables (suicidal ideation and attempts) and different independent variables. The variables with a p-value <0.25 during bivariable analysis were selected for the adjusted model and those variables having p value <0.05 were considered as statistically significant. Multicollinearity of the independent variables was tested using variance inflation factors (VIFs) and tolerance test, and there was no issue of multicollinearity. The values of tolerance for all variables were >0.2, and all values of VIF were <10.

### Ethical considerations

Before data collection, the aims of the research were explained to the study participants and written informed consent was obtained from each participant and from the legal guardians of participants for those unable to read and write after explaining the purpose and importance of the study before the interviews. Participation in the study was voluntary, and all information collected from the participants was kept confidential under the researcher's custody. Interviewers were trained to link participants found to be in physically risky conditions and/or in immediate need of counseling to psychologists and psychiatrists. This study was approved by the Institutional Health Research Ethics Review Committee of the College of Health and Medical Sciences of Haramaya University.

## Results

### Sociodemographic characteristic

A total 1,416 of sample size, 1,411 completed the interview with a response rate of 99.64%. Nearly half (52.2%) of the participants were female, and 36.4% of the respondents were in the aged group of 28–37. Almost all of the people, 84% were Muslim by religious. Likewise, more than half of the people (59%) were married, while 10.5% were divorced, and 3.3 were separated. Out Of the all participants, 34.9% were unable to read or write, as shown in Table 1.

### Substance-related factors

Of the total residents more than half (59%), 56.4% were used any kind of substance in their life time and in the past three months, respectively, as shown in Table 2.

### Prevalence of suicidal ideation and attempt among residents in Kersa, Haramaya and Harar health and demographic surveillance system, 2022

The total prevalence of suicidal ideation and attempts among the community residents of Kersa, Haramaya and Harar Health and Demographic Surveillance System were 9.8% [95% confidence interval (CI) (8.23–11.33)] and 6.2% [95% CI (4.97–7.50)], respectively.

### Factors associated with suicidal ideation among the Community of Kersa, Haramaya and Harar health and demographic surveillance system

Bivariate logistic regression analysis revealed that female sex, current and ever khat use, current alcohol consumption, cigarette smoking, history of mental illness, family history of mental illness and rural residence were significantly associated with suicidal ideation. However, in the multivariate logistic regression analysis,

**Table 1.** Socio-demographic and clinical characteristics of residents in Kersa, Haramaya and Harar Health and Demographic Surveillance System, 2022

| Variables | Category | Frequency (N = 1,411) | Percent (%) |
|---|---|---|---|
| Religion | Muslim | 1,185 | 84 |
| | Ortodox | 144 | 10.2 |
| | Protestant | 47 | 3.3 |
| | Catholic | 33 | 2.3 |
| | Others | 2 | 0.1 |
| Age. years | 18–27 | 442 | 31.3 |
| | 28–37 | 513 | 36.4 |
| | 38–47 | 279 | 19.8 |
| | >48 | 177 | 12.5 |
| Sex | Male | 675 | 47.8 |
| | Female | 736 | 52.2 |
| Marital status | Single | 294 | 20.8 |
| | Married | 833 | 59.0 |
| | Divorced | 148 | 10.5 |
| | Widowed | 90 | 6.4 |
| | Separated | 46 | 3.3 |
| Level of education | unable read and write | 492 | 34.9 |
| | Primary school | 204 | 14.5 |
| | Secondary school | 288 | 20.4 |
| | Diploma | 224 | 15.9 |
| | degree and more | 203 | 14.4 |
| Occupation | Unemployed | 182 | 12.9 |
| | self employed | 287 | 20.3 |
| | government employed | 351 | 24.9 |
| | Student | 115 | 8.2 |
| | Farmer | 120 | 8.5 |
| | house wife | 338 | 24.0 |
| | Others | 18 | 1.3 |
| Residence | Rural | 920 | 65.2 |
| | Urban | 491 | 34.8 |
| Living arrangement | with family | 1,302 | 92.3 |
| | Alone | 109 | 7.7 |
| Having history of mental illness | Yes | 259 | 18.4 |
| | No | 1,152 | 81.6 |
| History chronic medical illness | Yes | 181 | 12.8 |
| | No | 1,230 | 87.2 |

*Note*: private employee, NGO sector employee and religious leader.

history of mental illness and khat use were significantly associated with suicidal ideation.

In this study, the odds of having suicidal ideation among participants who had a previous history of mental illness was 6.82 times higher as compared to respondents who had history of mental

**Table 2.** Psychosocial, substance use related factors of residents in Kersa, Haramaya and Harar Health and Demographic Surveillance System, 2022

| Variables | Category | Frequency (N = 1,411) | Percent |
|---|---|---|---|
| Used any substance in life time | Yes | 832 | 59.0 |
| | No | 579 | 41.0 |
| Used any substance in the past 3 months | Yes | 796 | 56.4 |
| | No | 615 | 43.6 |
| Life time alcohol use | Yes | 144 | 10.2 |
| | No | 1,267 | 89.8 |
| Life time khat use | Yes | 783 | 55.5 |
| | No | 628 | 44.5 |
| Life time cigarette use | Yes | 1,137 | 80.6 |
| | No | 274 | 19.4 |
| Life time cannabis | Yes | 13 | 0.9 |
| | No | 1,398 | 99.1 |
| Cocaine in life time | Yes | 0.00 | 0.00 |
| | No | 1,411 | 100.0 |
| Current cigarette use | Yes | 255 | 18.5 |
| | No | 1,156 | 81.9 |
| Current alcohol use | Yes | 125 | 8.9 |
| | No | 1,286 | 91.1 |
| Current khat use | Yes | 746 | 52.9 |
| | No | 665 | 47.1 |
| Social support | Poor | 548 | 38.8 |
| | Moderate | 651 | 46.1 |
| | Strong | 212 | 15.1 |
| Stress full life event | Yes | 613 | 43.4 |
| | No | 798 | 56.6 |

illness [adjusted odds ratio (AOR) = 6.82 (4.63–10.05)] and the odds of suicidal ideation among respondents who had khat use was 3.24 times higher as that among respondents who had no khat use [AOR = 2.34 (1.48–3.69)], as shown in Table 3.

### Factors associated with suicidal attempt among the Community of Kersa, Haramaya and Harar health and demographic surveillance system

In bivariate logistic regression analysis, female sex, age > 48 years, current and ever-chewing khat, having a history of mental illness, previous history of mental illness and rural residence were significantly associated with suicide attempts. However, in the multivariate logistic regression analysis, history of mental illness, rural residence and khat use were significantly associated with suicide attempts.

In this study, the odds of having suicidal attempt among rural residents was about 4.32 times higher as compared to participants of urban residents [AOR = 4.32 (2.17–7.58)], and the odds of having suicide attempt among participants who had history of mental illness was 6.02 times higher as compared to respondents who had no history of mental illness [AOR = 6.02 (3.78–9.60)].

The odds of having suicide attempt among respondents who had history of khat use was 2.23 times higher as compared to respondents who had no history of khat use [AOR = 2.23 (1.29–3.85)], as shown in Table 4.

## Discussion

This community-based cross-sectional study aimed to assess the prevalence and associated factors of suicidal ideation and attempts among adults in Eastern Ethiopia. The findings revealed that 9.8% and 6.2% of the participants reported suicidal ideation and attempts, respectively. These rates are consistent with previous studies conducted in Ethiopia and other LMICs, which have reported varying prevalence rates of suicidal behavior due to differences in study settings, populations and methodologies (Gelaye et al., 2020; Tesfaye et al., 2021) The high prevalence of suicidal ideation and attempts in this study underscores the urgent need for targeted mental health interventions in the region.

The prevalence of suicidal ideation found in this study was higher than that reported in studies conducted in Northwest Ethiopia (Belete et al., 2021), South west Ethiopia (Tessema et al., 2024), the United Arab Emirates (Amiri, 2022) and China (Yang et al., 2015). The variation might be due to differences in the study setting, the tool used to assess suicidal behavior, design and sociocultural differences. In addition, the differences could be explained by the data collection method used.

The 12-month pooled prevalence of suicidal ideation was 9% (95% CI: 5–16%), and the lifetime prevalence of suicide attempts was approximately 4% (3–6%), according to a systematic review and meta-analysis of studies conducted in Ethiopia's general population(Bifftu et al., 2021). Using the Suicide Behavior Questionnaire-Revised, although slightly lower than our ideation prevalence, their figures are comparable to it and lower for attempts. Due to variations in sample, definition or reporting, our study may have found a higher attempt prevalence than many other studies.

The current study found that the prevalence of suicidal ideation (9.8%) was lower than that reported in another Ethiopian study, which reported a prevalence of 21.9% (95% CI: 18.40–25.20) (Tadesse et al., 2021). This discrepancy may be attributed to differences in study settings, sociocultural factors and the tools used to assess suicidal behavior.

The prevalence of suicide attempts in this study was 6.2% (95% CI: 4.97–7.50), which is higher than rates reported in previous studies conducted in Ethiopia (3.8%) (Fekadu et al., 2016), Nigeria (0.7%) (Gureje et al., 2011), Zambia (2.3%) (Pengpid and Peltzer, 2021) and South Africa (2.9%) (Joe, 2008). This higher prevalence could be due to several factors: 1) the phrasing of the question used in the interview, which asked, "Have you ever considered ending your life?" rather than using the term "suicide," potentially making participants more willing to disclose suicidal thoughts and 2) the inclusion of a broader age range in this study, as younger individuals are known to report higher rates of suicidal thoughts and behaviors. Additionally, the lifetime prevalence of suicide attempts in this study was higher than the global estimate of 2.7% (Nock et al., 2008), which may reflect regional variations in risk factors and cultural attitudes toward suicide.

However, the prevalence of suicide attempts in this study was lower than that reported in another Ethiopian study, which found a prevalence of 13.1% (95% CI: 10.60–15.80) (Tadesse et al., 2021). This variation may be explained by differences in study settings,

**Table 3.** Factors associated with suicidal ideation among Residents of Kersa, Haramaya and Harar Health and Demographic Surveillance System, Eastern Ethiopia, 2022

| Variables | Suicidal ideation | | COR (95% CI) | AOR (95% CI) |
|---|---|---|---|---|
| | Yes | No | | |
| **Sex** | | | | |
| Male | 85 | 590 | 1 | 1 |
| Female | 53 | 683 | 0.53 (0.37–0.77) | 0.86 (0.57–1.30) |
| **Marital status** | | | | |
| Married | 71 | 762 | 1 | 1 |
| Single | 37 | 257 | 0.64 (0.42–0.98) | 0.78 (0.49–1.24) |
| Divorced/separated/widowed | 30 | 254 | 0.82 (0.49–1.36) | 0.93 (0.54–1.63) |
| **Khat use (current, life time)** | | | | |
| Yes | 107 | 690 | 2.91 (1.92–4.41) | 2.34 (1.48–3.69) |
| No | 31 | 583 | 1 | 1 |
| **Having history of mental illness** | | | | |
| Yes | 181 | 88 | 7.84 (5.40–11.37) | 6.82 (4.63–10.05) |
| No | 60 | 1,092 | 1 | 1 |
| **Having chronic medical illness** | | | | |
| Yes | 30 | 151 | 2.06 (1.33–3.20) | 0.62 (0.33–1.16) |
| No | 108 | 1,122 | 1 | 1 |
| **Current substance users** | | | | |
| Yes | 107 | 689 | 2.93 (1.93–4.42) | 1.45 (0.21–9.7) |
| No | 31 | 584 | 1 | 1 |
| **Cigarette use** | | | | |
| Yes | 51 | 223 | 2.76 (1.89–4.01) | 2.57 (0.74–3.93) |
| No | 138 | 1,050 | 1 | 1 |

Abbreviations: AOR, adjusted odds ratio; COR, crude odds ratio.

sociocultural contexts and the methodologies used to assess suicidal behavior. These findings highlight the importance of considering contextual and methodological factors when interpreting and comparing suicide-related data across studies. According to a recent study, the prevalence of suicide behavior was much higher among residents of Dessie town, Northeast Ethiopia, which was devastated by the war: 15.3% (95% CI: 12.5%–18.3%) (Birhan et al., 2025). Comparing this to our results, we found that our ideation rate (9.8%) is higher than many general population studies but lower than in some conflict or high-stress settings; our attempt prevalence (6.2%) is slightly lower than in the war-affected Dessie study and falls between the general population pooled estimate (≈9%).

The study identified several factors significantly associated with suicidal ideation and attempts. A history of mental illness was strongly associated with both suicidal ideation (AOR = 6.82, 95% CI: 4.63–10.05) and suicide attempts (AOR = 6.02, 95% CI: 3.78–9.60). This finding aligns with global evidence indicating that individuals with mental health disorders, such as depression, anxiety and bipolar disorder, are at a significantly higher risk of suicidal behavior (Franklin et al., 2017). The strong association between mental illness and suicidal behavior highlights the critical need for early identification, diagnosis and treatment of mental health conditions in this population. Strengthening mental health services and integrating them into primary healthcare systems could play a

pivotal role in reducing the burden of suicidal behavior in Eastern Ethiopia.

Khat use was another factor significantly associated with both suicidal ideation (AOR = 2.34, 95% CI: 1.48–3.69) and suicide attempts (AOR = 2.23, 95% CI: 1.29–3.85). Khat chewing is a common cultural practice in Eastern Ethiopia and has been linked to various mental health issues, including psychosis, depression and anxiety (Hailemariam et al., 2015). The psychoactive properties of khat, particularly its stimulant effects, may exacerbate underlying mental health conditions and contribute to impulsive behaviors, including suicidal acts (Mars et al., 2014). Public health interventions aimed at reducing khat use, coupled with community education on its mental health consequences, could help mitigate the risk of suicidal behavior in this population.

Rural residence was significantly associated with suicide attempts (AOR = 4.32, 95% CI: 2.17–7.58). This finding is consistent with studies from other LMICs, which have reported higher rates of suicidal behavior in rural areas due to limited access to mental health services, higher levels of poverty and social isolation (WHO, 2021). The lack of mental health infrastructure in rural settings often results in untreated mental health conditions, increasing the risk of suicidal behavior. Addressing these disparities requires targeted investments in rural mental health services, including training healthcare workers, establishing community-

**Table 4.** Factors associated suicide attempt among Residents of Kersa, Haramaya and Harar Health and Demographic Surveillance System, Eastern Ethiopia, 2022

| Variables | Suicide attempt | | COR (95% CI) | AOR (95% CI) |
|---|---|---|---|---|
| | Yes | No | | |
| **Sex** | | | | |
| Male | 52 | 623 | 1 | 1 |
| Female | 36 | 700 | 1.62 (1.05–2.52) | 0.89 (0.54–1.45) |
| **Age, years** | | | | |
| 18–27 | 28 | 414 | 1 | 1 |
| 28–37 | 30 | 483 | 0.92 (0.54–1.56) | 0.80 (0.50–1.29) |
| 38–47 | 10 | 269 | 0.55 (0.26–1.15) | 0.69 (0.39–1.22) |
| >48 | 20 | 157 | 1.88 (1.03–3.44) | 0.85 (0.46–1.58) |
| **Area of residence** | | | | |
| Rural | 78 | 842 | 4.46 (2.28–8.68) | 4.32 (2.17–7.58) |
| Urban | 10 | 481 | 1 | 1 |
| **History of mental illness** | | | | |
| Yes | 48 | 211 | 6.32 (4.05–9.86) | 6.02 (3.78–9.60) |
| No | 40 | 1,112 | 1 | 1 |
| **Khat use** | | | | |
| Yes | 65 | 732 | 2.28 (1.40–3.71) | 2.23 (1.29–3.85) |
| No | 33 | 591 | 1 | 1 |

Abbreviations: AOR, adjusted odds ratio; COR, crude odds ratio.

based mental health programs and raising awareness about mental health issues.

The high prevalence of suicidal ideation and attempts in this study (nearly one in 10 individuals) highlights the urgent need for comprehensive mental health interventions in Eastern Ethiopia. Suicidal behavior is a multifaceted issue influenced by a combination of psychological, social and environmental factors. As such, interventions should adopt a holistic approach that addresses the underlying determinants of suicidal behavior, including mental illness, substance use and socioeconomic challenges. Community-based mental health programs, coupled with efforts to reduce stigma and improve access to care, could play a critical role in preventing suicidal behavior in this population.

This study has several limitations. First, the cross-sectional design limits the ability to establish causal relationships between the identified risk factors and suicidal behavior. Second, the reliance on self-reported data may introduce recall bias, particularly for sensitive topics such as suicidal ideation and attempts. Future longitudinal studies are needed to explore factors and their temporal relationships with suicidal behavior.

In conclusion, this study highlighted the high prevalence of suicidal ideation and attempts among adults in Eastern Ethiopia and identified key risk factors, including a history of mental illness, khat use and rural residence. These findings underscore the need for targeted mental health interventions that address the unique challenges faced by this population. Strengthening mental health services, reducing substance use and addressing socioeconomic disparities are critical steps toward reducing the burden of suicidal behavior in Eastern Ethiopia.

**Open peer review.** To view the open peer review materials for this article, please visit http://doi.org/10.1017/gmh.2025.10113.

**Data availability statement.** To protect the anonymity of the participants, the data generated and analyzed during the current study are not publicly available. Upon reasonable request, materials were obtained from the corresponding author.

**Acknowledgements.** The authors would like to extend our appreciation to Haramaya University for providing the technical and financial support for this study. The authors would also like to thank all respondents who participated in this study and their commitment to responding to our questions.

**Author contribution.** All authors made substantial contributions in planning, designed the study, implemented the study, conducted statistical analysis, interpreted the data, conceived the idea, wrote the proposal and participated in data collection, analysis and writing of the paper. Additionally, all authors participated in drafting and revising the paper. All authors contributed to revising the manuscript for important content, and read and approved the final version of the manuscript.

**Competing interests.** The authors declare none.

**Ethical approval and consent to participate.** Ethical clearance was obtained from the Institutional Health Research Ethics Review Committee of the College of Health and Medical Sciences of Haramaya University. All methods were performed in accordance with the relevant guidelines and regulations. Then, data collection was initiated after a letter of the corporation that was obtained from Haramaya University College of Health and Medical Sciences to each Woreda and Kebele administrators. Official permission was secured from Woreda and Kebele administrators. Informed consent was obtained from each participant and from the legal guardians of participants who were unable to read and write after explaining the purpose and importance of the study before the interviews. Participation in the study was voluntary, and all information collected from the participants was kept confidential under the researcher's custody. Interviewers were trained to link participants found to be in physically risky conditions and/or in immediate need of counseling to psychologists and psychiatrists. To ensure the safety of data collectors and participants from the COVID-19 pandemic diseases, training was given to data collectors were trained on the proper use of coronavirus requisition measures and to protect the participants from the pandemic.

**Informed consent.** Participation was voluntary and information was collected anonymously after obtaining voluntary written informed consent from each respondent by assuring confidentiality throughout data collection period. For those unable to read and write, written informed consent were taken from their legal guardians. Participants were told the objective of the study and their right to refuse or answer the questionnaires and were given the right to stop or withdraw at any point of time during data collection. Confidentiality was maintained by omitting their name and personal identification.

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
