## [Reviewer Report]

Thank you for the submission of this manuscript for review!

The manuscript reports on the prevalence and associated factors of suicidal ideation and attempts among adults living in three surveillance sites in Eastern Ethiopia. The authors conducted a cross-sectional survey of 1,411 adults and used logistic regression to identify associated factors, including rural residence, lifetime khat use, and prior history of mental illness.

The paper has a number of strengths, with an interesting topic of suicide in LMICs that remains under-researched. The manuscript’s use of community-based surveillance sites and validated tools (WHO-CIDI, ASSIST, Oslo Social Support Scale) is useful for public health implementation and potential future comparative research. In addition, the paper emphasizes structural barriers in Ethiopia such as rurality and stigma, which are important factors related to mental health outcomes.

However, the manuscripts needs major revision to demonstrate its contribution to new knowledge in this area and requires greater clarity and rigor in its methods and interpretation before it could be considered for publication.

Areas for Revision:

- Identification of New Insights and Justification for the Study

o The authors state in the abstract and introduction that “little is known about suicidal behavior in these nations, especially Ethiopia” (p. 3), yet numerous studies from various regions of Ethiopia have already reported on suicidal ideation and attempts using similar designs and factors—such as khat use, rural residence, and history of mental illness. The discussion itself acknowledges this overlap both for suicidal ideation and suicide attempts by citing numerous studies from Ethiopia and other countries.

o The manuscript must be edited to explain how this study adds new information to the existing literature on suicidal behavior in Ethiopia. Without a clear rationale for focusing on Kersa, Haramaya, and Harar HDSS sites—or how findings differ from existing data—I am left wondering what new insights this work brings. The manuscript should explicitly address why this region deserves separate attention, particularly given similar demographic and cultural profiles to other previously studied areas.

- Methodological Explanation and Depth

o The criteria for variable inclusion in the multivariable model are unclear. Was a p-value cutoff used in the bivariable analysis? Were theoretical considerations used? Please clarify this for reproducibility.

o While the authors mention checking for multicollinearity using VIF, they do not report the actual VIF values or clarify whether any variables were removed as a result. Please include the chosen VIF threshold for multicollinearity and any actions taken, as well as listing which variables if any were above the threshold.

o The study does not include any stratified or interaction analysis to deepen understanding of how factors like khat use, social support, or rurality may operate in combination. There is also no subgroup analysis (gender, age, mental health comorbidity) that would add depth to the results. Please consider adding further analysis on subgroups to enhance the findings’ additions to the field.

- Clarity in Description of Results:

o Overall, the results section would benefit from greater clarity around response rates, missing data, and any exclusions made during analysis.

o Throughout the abstract and results sections, the authors refer to factors as “predictors” (e.g., “Suicidal thoughts were significantly predicted by lifetime khat usage…”, p. 3), although this is a cross-sectional study that cannot establish temporal order. The framing should be revised to focus on associations, and the limitations section should explicitly discuss the inability to infer causality.

o The manuscript inconsistently refers to “lifetime” suicidal ideation and attempts, but this is not always specified in tables or methods. The distinction between past-year vs. lifetime prevalence should be clearly stated.

- Increased Contextualization in Discussion:

o The discussion section would benefit from a more critical reflection on why this region may or may not have similar findings to other areas in Ethiopia. The association with khat, for example, is often cited in Ethiopian research—what does this study uniquely add to understanding the role of khat in suicidal behavior?

o Additionally, the AOR for suicidal ideation with khat use is reported as “AOR = 3.24 (95% CI: 2.19–17.6)” (p. 12), which is a very wide range, suggesting model instability or sparse data for this subgroup. This issue should be discussed in the limitations, as it weakens confidence in the robustness of the findings. Additional limitations related to the limits of the cross-sectional study methodology, study population, and other unique study aspects should also be addressed more clearly.

- Writing/Editing: The manuscript requires careful language editing to improve grammar and reduce redundancy. There are many repeated and awkward phrases as well as grammatical errors (e.g. “suicidal attempts” instead of “suicide attempts”) that distract from the overall manuscript.

---

## [Reviewer Report]

Review of Manuscript Entitled “ SUICIDAL IDEATION AND ATTEMPTS AMONG ADULTS IN EASTERN ETHIOPIA: A COMMUNITY-BASED CROSS-SECTIONAL STUDY”.

This paper aims to assess the frequency and contributing factors of suicide ideation and attempts among adults in the Eastern Ethiopian.

Major Concerns

I have the following suggestions:

- Abstract: I would remove the sentence about the software used; it is not relevant.

- The Impact Statement section is missing

- Introduction: The section is clearly presented and effectively communicates the purpose of the research

- Method and Material:

1. I would change the section title to simply ‘Methods’.

2. I understand that these are three regions in eastern Ethiopia, but I have some doubts since only the geographic distribution of two sites, Harar HDSS and Kersa HDSS, is explained. I believe it would help clarify their geographic distribution if this point were explained a bit more clearly.

3. Sample size selection. Please provide a more detailed explanation of the sample size determination process, including the population size used for the calculations to obtain the sample (n=1416).

4. I couldn’t find Figure 1; perhaps it includes the process of participant selection.

5. Please review this section regarding the sampling techniques used to obtain the sample, as it is not very clear.

6. There are two repeated paragraphs:

…” a design effect of 1.5………” (líneas 25-28, pág 6).

“…..a multiple-stage sampling technique was used…..” (linea 30-34, pág 6)

- Data collection tools and measurements

1. The dependent variables are suicidal ideation and suicide attempts, so they should be placed at the beginning of this section, immediately after ‘...were employed to collect all essential data’

2. “Is there any temporal information available for this variable? For instance, whether the individual has attempted suicide within the past six months?” If such information is available, it would be highly valuable to include it in the analysis.

3. Socio-demographic variables: I would include all the variables that were analyzed and appear in Table 1, e.g., age groups, sex, marital status, etc."

4. Use of psychoactive substances, social support, and stressful life events screening: I would include, for each of these, the variables or categories that comprise them. For example, use of psychoactive substances: used any substances in lifetime? (Yes/No); used any substances in the past 3 months? (Yes/No)...

5. Statistical analysis: Please provide a clearer explanation of how the logistic regression models were conducted, specifying the dependent variable and the independent variables

- Results

1. The variable ‘religion’ is not included in Table 1, although it is mentioned in the text. I would include it in the table

2. “…. 12.9% of all participants were unable to read or write…”. This figure does not match the information in the table, which indicates 34.9%.

3. Table 1: I would combine the categories ‘separated’ and ‘divorced’ (194, 13.8%) to increase statistical power in the regression models."

4. Table 1: I recommend using more specific labels for the ‘educational level’ variable to enhance clarity. Please clarify which educational levels grades 1–8 and 9–12 correspond to in the Ethiopian context. Do they represent primary and secondary education, respectively?

5. On page 12, it states: ….”the odds of having suicidal ideation among respondents who were female were about 1.69 times higher than those who were male (AOR=1.69 (1.62-1.57))...' However, the table indicates AOR: 0.99 (0.62, 1.57). Which is correct?"

6. Tables 3 and 4: I would include the number of cases for suicidal ideation and suicide attempts (yes and no)

7. Tables 3 and 4: I would remove the p-value. There should be two, one for the crude models and another for the adjusted models. However, they are not necessary since the confidence intervals (CIs) allow for interpretation

8. Tables 3 and 4: Why do the adjustment variables in the logistic regression models differ between suicidal ideation and suicide attempts?

9. The adjusted logistic regression models include repeated variables. For example, current cigarette use and lifetime cigarette use. Similar variables should not be included in the same model as this causes collinearity. I recommend including only one of them. The same applies to the use of khat.

10. Age groups should also be included in the regression models, as it is well established that age significantly influences suicidal ideation and attempts. I recommend that you consider creating age groups comparable to those used in previous studies, so that age-related prevalences can be compared in the discussion section

11. In my view, incorporating social support and stressors into the regression models would provide a more comprehensive and informative analysis

12. Table 4: There seems to be an error in the variable ‘sex,’ as the figures are exactly the same as those in Table 3.

Discussion:

I recommend that the authors review these recent articles on suicidal ideation and attempts in Ethiopia, as they are not currently included in the references.

1. Bifftu BB, Tiruneh BT, Dachew BA, Guracho YD.Prevalence of suicidal ideation and attempted suicide in the general population of Ethiopia: a systematic review and meta-analysis. Int J Ment Health Syst. 2021 Mar 24;15(1):27. doi: 10.1186/s13033-021-00449-z.

2. Tessema SA, Torba AN, Tesfaye E, Alemu B, Oblath R. Suicidal behaviours and associated factors among residents of Jimma Town, Southwest Ethiopia: a community-based cross-sectional study. BMJ Open. 2024 Sep 24;14(9):e085810. doi: 10.1136/bmjopen-2024-085810.PMID: 39317502

3. Birhan Z, Shegaw M, Kunno K, Anbesaw T, Woretaw L, Tedla A, Munie BM, Belete A.Front Psychiatry. Prevalence of suicidal behavior and its associated factors among individuals living in war-affected areas of Dessie Town, northeast Ethiopia, in 2022: a cross-sectional study.2025 May 29;16:1453526. doi: 10.3389/fpsyt.2025.1453526. eCollection 2025.

I consider this to be an interesting study, with the potential to provide scientific evidence on the prevalence of suicidal ideation and suicide attempts in the eastern region of Ethiopia. The major revisions needed are primarily focused on the methodology and results sections. I consider the following aspects essential: (1) A summary of the population characteristics in the three regions analyzed, including the population size of each, should be provided; (2) offering a more detailed description of the sampling techniques used to obtain the 1,411 surveys; (3) thoroughly describing all variables analyzed in the methodology section; and (4) reviewing the selection of variables included in the logistic regression models.

I encourage the authors to take these comments into account to enhance the methodological quality of their manuscript.

---

## [Reviewer Report]

Thank you for your succinct response to many of my comments from the first draft of your manuscript. I very much appreciate the additions about the Eastern Ethiopian region in particular and clarification of methods and results based on your approach.

I believe that this study should be accepted pending grammatical review from the copy editor and minor revisions based on continued queries from my prior comments:

o The study does not include any stratified or interaction analysis to deepen understanding of how factors like khat use, social support, or rurality may operate in combination. There is also no subgroup analysis (gender, age, mental health comorbidity) that would add depth to the results. Please consider adding further analysis on subgroups to enhance the findings’ additions to the field.

- Update: I understand that you did not choose to add further sub-groups. Perhaps you could add this to the discussion for further analysis in another study.

o The manuscript inconsistently refers to “lifetime” suicidal ideation and attempts, but this is not always specified in tables or methods. The distinction between past-year vs. lifetime prevalence should be clearly stated.

- Update: You deleted “lifetime” throughout the manuscript but did not clarify the definition of suicidal ideation. How did you define suicidal ideation with the study subjects—within a past time frame? lifetime? Please clarify in methods.

o Additionally, the AOR for suicidal ideation with khat use is reported as “AOR = 3.24 (95% CI: 2.19–17.6)” (p. 12), which is a very wide range, suggesting model instability or sparse data for this subgroup. This issue should be discussed in the limitations, as it weakens confidence in the robustness of the findings. Additional limitations related to the limits of the cross-sectional study methodology, study population, and other unique study aspects should also be addressed more clearly.

- Update: Please add to the limitations section to complete a full paragraph with limitations. Please provide more clarity on specific limitations of this study, beyond general study design, including the limitations addressed in the previous comment. This clarifies that you understand the limits of your study and the potential areas of growth for future research, which should also be included in the discussion section.

- Update: I see that you updated your methods section to describe that you included values with p<0.25. from the bivariate regression to add to the adjusted model. Since you are removing the p-values from the AOR per the tables, please list which variables were included vs removed for the adjusted analyses for both suicidal ideation and attempt either in the methods or results along with each p-value for reference.

---

## [Reviewer Report]

Although the authors have improved the article, there are still important points that should be reviewed:

In the section on Data Collection Instruments and Measurements:

- Review the wording of the first sentences as some words are repeated.

E.g., 'face-to-face interviews.

- In the sentence “the socio-demographic and economic characteristics of the participants”. Include the term ‘clinical,’ as variables such as: previous history of mental illness, history of chronic medical illness, etc., are studied.

- As I mentioned in the previous review, I believe it is necessary to clarify which are the dependent variables of this study (suicidal ideation and attempts). This can be indicated here or in the statistical analysis section

In the section on Statistical analysis

- sentence: “Descriptive statistical tests were used to provide clear distribution of the data”.

What do they mean by this sentence? There are no quantitative variables, so they couldn’t have applied tests in the descriptive analysis. If it’s not clear, I would remove it.

- Paragraph: “Multicollinearity of the independent variables was tested using variance inflation factors (VIF) and tolerance test and there was no issue of multicollinearity. The values of tolerance for all variables were greater than 0.2 and all values of VIF were less than 10”

I would like the authors to explain why they applied the VIF and tolerance test and where the results are reported.

In the section on Results.

Table 1 describes the collected socio-demographic and clinical variables, while Table 2 presents psychosocial and substance use variables.

- Add ‘and clinical’ to the title of Table 1

The multivariable model with many variables, especially those with many categories, makes the coefficient estimates unstable, as seen in the confidence intervals, which become very wide.

- I recommend that variables with many categories included in the regression models be recategorized into fewer categories.

For example: The occupation variable (Unemployed, self-employed, government-employed, student, farmer, housewife, others) has categories with few cases.

Similarly, for marital status, with 5 categories, the results of the bivariate analysis in Table 3 show the category ‘Separated’ with only 4 cases of suicidal ideation, 42 without ideation, and a COR of 2.81 (0.89-8.819). The variable can be recoded into 3 categories: (1) single (2) Married and (3) Separated/ Divorced/Widowed.

- Both current use and lifetime use of certain substances have been incorporated into the models, leading to collinearity.

For example, as seen in Table 2, there are 783 individuals who have used Khat at least once in their lifetime, and another variable with 746 individuals who currently use Khat. The second variable is inherently included within the first.

It would be advisable to create a variable that captures Khat use, such as: use of Khat (current, former user, never), or to include only one of the two in the analyses (current or lifetime use). The same applies to the other substances.

In the section on Discussion:

I recommend that the authors review these recent articles on suicidal ideation and attempts in Ethiopia, as I believe they could be useful for this section.

1. Bifftu BB, Tiruneh BT, Dachew BA, Guracho YD.Prevalence of suicidal ideation and attempted suicide in the general population of Ethiopia: a systematic review and meta-analysis. Int J Ment Health Syst. 2021 Mar 24;15(1):27. doi: 10.1186/s13033-021-00449-z.

2. Tessema SA, Torba AN, Tesfaye E, Alemu B, Oblath R. Suicidal behaviours and associated factors among residents of Jimma Town, Southwest Ethiopia: a community-based cross-sectional study. BMJ Open. 2024 Sep 24;14(9):e085810. doi: 10.1136/bmjopen-2024-085810.PMID: 39317502

3. Birhan Z, Shegaw M, Kunno K, Anbesaw T, Woretaw L, Tedla A, Munie BM, Belete A.Front Psychiatry. Prevalence of suicidal behavior and its associated factors among individuals living in war-affected areas of Dessie Town, northeast Ethiopia, in 2022: a cross-sectional study.2025 May 29;16:1453526. doi: 10.3389/fpsyt.2025.1453526. eCollection 2025.

I hope these comments are useful to you.

---

## [Reviewer Report]

Thank you for all of your work updating this manuscript according to many recommendations in prior rounds of review. I have one minor grammatical revision which is that on page 40 line 30, you state “variables such as previous history of mental illness, and lifetime khat use” when referencing variables that were significantly associated with suicidal ideation in the multivariate logistic regression. In fact, these are the only two variables that are significant, so the “such as” should be removed as it assumes that these are two of a larger group of variables. I would recommend reviewing the manuscript for other instances of this unclear language, however this is very small and I would suggest accepting the manuscript with its current revisions. Thank you for the pleasure of working on this review.

---

## [Editor Report]

The authors have not addressed very relevant concerns voiced by reviewers in their resubmission.

If the authors cannot address these concerns, we cannot accept the manuscript.

See below for the concerns:

The authors omit the issue raised regarding the analysis of the study.

“The multivariable model with many variables, especially those with many categories, makes the coefficient estimates unstable, as seen in the confidence intervals, which become very wide.

- I recommend that variables with many categories included in the regression models be recategorized into fewer categories.

For example: The occupation variable (Unemployed, self-employed, government-employed, student, farmer, housewife, others) has categories with few cases.

Similarly, for marital status, with 5 categories, the results of the bivariate analysis in Table 3 show the category ‘Separated’ with only 4 cases of suicidal ideation, 42 without ideation, and a COR of 2.81 (0.89-8.819). The variable can be recoded into 3 categories: (1) single (2) Married and (3) Separated/ Divorced/Widowed.

- Both current use and lifetime use of certain substances have been incorporated into the models, leading to collinearity.

For example, as seen in Table 2, there are 783 individuals who have used Khat at least once in their lifetime, and another variable with 746 individuals who currently use Khat. The second variable is inherently included within the first.

It would be advisable to create a variable that captures Khat use, such as: use of Khat (current, former user, never), or to include only one of the two in the analyses (current or lifetime use). The same applies to the other substances”